# Examination of Countermovement Jump Performance Changes in Collegiate Female Volleyball in Fatigued Conditions

**DOI:** 10.3390/jfmk8030137

**Published:** 2023-09-17

**Authors:** Paul T. Donahue, Ayden K. McInnis, Madelyn K. Williams, Josey White

**Affiliations:** School of Kinesiology and Nutrition, University of Southern Mississippi, Hattiesburg, MS 39406, USAmadelyn.k.williams@usm.edu (M.K.W.); josey.white@usm.edu (J.W.)

**Keywords:** countermovement jump, athlete monitoring, force–time curve

## Abstract

The purpose of this investigation was to examine changes in countermovement vertical jump performance after a single sport-specific training session in a sample of collegiate female volleyball athletes. Eleven NCAA Division I volleyball athletes performed countermovement vertical jumps with and without an arm swing prior to and immediately after a sport-specific training session. Each participant completed two jumps in each condition using a portable force platform. Paired samples t-tests were performed within each jump condition. When using an arm swing, mean braking force was the only variable to display a statistically significant change (*p* < 0.05). In the no-arm-swing condition, mean propulsive force, propulsive net impulse, jump height and reactive strength index modified all statistically increased (*p* < 0.05). Time to takeoff was statistically reduced (*p* < 0.05). Additionally, a single-subject analysis was performed across all eleven participants resulting in general trends seen in the no-arm-swing condition, whereas the arm-swing condition displayed inconsistent findings across participants.

## 1. Introduction

The sport of volleyball emphasizes having a strong ability to jump as it is a critical component of the technical skills needed to compete (blocking and hitting) [1,2,3]. Vertical jump testing has become a common assessment of neuromuscular fatigue in athletic populations [4,5]. This is in part due to the ease of testing protocols and insight obtained from specific variables that relate directly to neuromuscular function. Jump assessments have been performed using a multitude of methodologies making jump height and peak power common variables of interest [6]. This has created conflicting findings throughout the literature regarding changes in jump performance in states of neuromuscular fatigue for a variety of reasons. 

First, several protocols have been used when performing vertical jump assessments. The most common difference between these protocols is the utilization of an arm swing (AS) movement. Previous investigations have shown that, when using an arm swing, jump performance will typically be greater, as evidenced by larger jump heights being achieved [7,8,9,10,11]. It was been proposed that, when using an AS during vertical jump assessments, individuals may use a different movement strategy [7]. While the AS may provide a level of ecological validity to the testing, any fatigue an individual may be experiencing can be masked by changing the relative usage of the arms to create upward momentum. Second, the device being used to assess jump performance can have a large impact on our ability to determine the level of fatigue. When using traditional field-testing devices (Vertec), jump height is the only variable that can be collected. As mentioned previously, the usage of an AS can mask changes in jump strategy to maintain jump height. When using these traditional devices, an AS jump is required to determine jump height. Similarly, when using jump mats, only jump height can be assessed. Though either an AS or no arm swing (NAS) methodology can be used, underlying strategies that determine jump height and offer a more thorough analysis cannot be assessed. 

Cormack et al. reported [12] reductions in jump performance immediately post-match that were maintained for 24 post-match from pre-match in Australian rules football athletes. Specifically, flight time was reduced by approximately 3%, and relative mean force was reduced by approximately 2%. In contrast, Hoffman et al. [13] found peak power and force were maintained pre- to post-match in soccer athletes. Interestingly, Johnston et al. [14] reported no changes in peak force while reductions in peak power were present over a competition period. This reduction in power with no reduction in force points to a change in movement velocity rather than force outputs in a fatigued state [14]. More recently it has been reported that no changes were seen in male volleyball athletes from pre- to post-sport-specific training sessions [15]. Thus, this investigation sought to examine the changes in vertical jump assessments using both AS and NAS conditions in a sample of female collegiate volleyball athletes with pre- and post-sport-specific training. 

## 2. Materials and Methods

This investigation employed a cross-sectional study design to assess changes in CMJ performance before and after a sport-specific volleyball training session. Testing took place during the spring training period and was a part of the regular athlete monitoring program that all athletes participated in as a part of their sports participation. The training session was approximately 2 h in duration. During the training session, six participants wore inertial sensors (Vert, Mayfonk Athletic, Fort Lauderdale, FL, USA) to measure jump counts. The average jump count for the six participants during the session was 103.66 with a range of 81 to 165 jumps. 

### 2.1. Subjects

Eleven NCAA Division I female indoor volleyball athletes (age: 19.77 ± 1.09 years; height: 178.56 ± 7.81 cm; body mass: 72.42 ± 7.81 kg) participated in this study. A post-hoc power analysis was performed using G*Power (version 3.1.9.7). This calculation was completed using the jump height from the no-arm-swing condition (NAS) with an effect size of 1.76. Observed power was calculated as 0.99. All participants were cleared to partake in team-related activities by the sports medicine staff and were free of injury at the time of testing and during the 4 weeks before testing taking place. This study was conducted according to the guidelines of the Declaration of Helsinki and was approved by the University of Southern Mississippi institutional review board (20-478). Each participant provided informed written consent prior to testing.

### 2.2. Procedures

Participants performed all jumping trials after performing a warm-up directed by the team’s strength and conditioning staff. Warm-ups took approximately 10 min to complete and consisted of dynamic lower body movements as well as submaximal vertical jumps. All trials were completed using a self-selected countermovement depth and foot position. Verbal instructions were given before initiation of each trial to “jump as high as possible”. 

During the NAS trials, a dowel (polyvinyl chloride, <1.0 kg) was placed across the upper back in a manner similar to the position of a barbell during the back squat exercise [1,2]. Participants were instructed to maintain contact between the dowel and the upper back during the duration of the trial. During arm swing (AS) trials, participants were instructed to begin each trial with both arms raised above their head. They were then allowed to swing their arms in any manner they desired to obtain the greatest jump height. All trials were collected using a portable force platform (AMTI, Accupower, Watertown, MA, USA) sampling at 1000 Hz. Each trial began with participants having one second of quiet standing before being given a “3, 2, 1, Go” countdown. During the quiet standing phase, body mass was calculated from the vertical ground reaction force. A 30-s rest period was given between trials. NAS trials were performed before AS trials during both testing sessions. 

### 2.3. Data Analysis

Raw vertical ground reaction force data was then exported and analyzed using a customized Excel spreadsheet (v.2308, Microsoft, Redmond, WA, USA) [1,2,16]. The spreadsheet was modeled using methods previously reported by Chavda et al. [17]. CMJ phase definitions followed those suggested by McMahon et al. [18]. Briefly, the phases of interest for this investigation were defined as the braking and propulsive phases. Braking was defined as the point at which vertical ground reaction force surpassed the calculated body mass during one second of quiet stance prior to the trial initiation until the instant the center of mass velocity reaches zero. The propulsive phase was defined as the end of the braking phase to the point of takeoff. The center of mass velocity was calculated by finding the center of mass acceleration for each sample by subtracting the calculated body mass from the vertical force data. Then, integration of acceleration data with respect to time using the trapezoidal rule, beginning 30 ms before movement initiation as recommended by Owen et al. [19], provided the center of mass velocity. Integration of the center of mass velocity data with respect to time provided the center of mass displacement. As for variable calculations, time to takeoff was calculated as the duration from movement initiation to the point of takeoff. Reactive strength index modified was calculated as jump height divided by time to takeoff [20]. Finally, all force variables are presented as net force (measured force – body mass). 

### 2.4. Statistical Analysis

Mean data for the two trials in each condition were used in the statistical analysis. Reliability analysis for each variable used both intraclass correlation coefficient (ICC) and coefficient of variation (CV) from the pre-testing data. ICC was calculated using a two-way random approach. Reliability was deemed acceptable with ICC values greater than 0.80 and CV values of less than 10%. To compare conditions, a paired samples t-test was conducted for each variable. Significance for all tests was a priori set at *p* < 0.05. Effect sizes were calculated as Hedge’s *g* and interpreted using the criteria of trivial (<0.2), small (0.2–0.6), moderate (0.61–1.20), large (1.21–2.0), very large (2.0–4.0) and nearly perfect (≥4.0) [21]. All statistical analyses were performed using SPSS (v28.0, SPSS Inc., Chicago, IL, USA). 

Additionally, single-subject analyses were performed on each variable of interest to determine if the changes seen were outside the individual variability exhibited during the pretest. Variability was assessed using pretest CV values [22].

## 3. Results

All variables demonstrated acceptable levels of reliability (Table 1). Data are reported as means ± SD and displayed in Table 2. In the AS condition, only mean braking force displayed a significant increase from pre to post (*p* = 0.047, *g* = 0.66). In the NAS condition, mean propulsive force increased from pre to post (*p* = 0.002, *g* = 1.18) coinciding with an increase in propulsive net impulse (*p* = 0.038, *g* = 0.70). Jump height significantly improved pre to post (*p* = 0.001, *g* = 1.70). Additionally, time to takeoff was significantly reduced (*p* = 0.015, *g* = 0.85). Finally, RSIm was significantly improved (*p* = 0.001, *g* = 1.47).

When using the single subject analysis, each variable displayed an individual response, where both positive and negative changes were seen as well as no change. In the AS condition, seven participants showed an increase in mean braking force with two having a reduction and two with no change. Three participants showed a reduction in braking duration, while two had an increase in duration and six had no change. Braking net impulse was increased in six individuals, with decreases in two individuals and no change was shown in three. Propulsive mean force was increased in four, reduced in four, and showed no change in three. Propulsive duration increased in four, reduced in three, and no change was seen in four. Propulsive net impulse was increased in four participants, reduced in four, and showed no change in three. Five participants displayed an increase in countermovement depth, with one reducing depth and five having no change. Jump height was increased in six, decreased in four, and no change was seen in one participant. Time to takeoff was reduced in five individuals, increased in four and no change was seen in two. Lastly, RSIm was increased in five participants, reduced in two individuals and no change was seen in four.

During the NAS condition, six participants showed an increase in mean braking force. Two displayed a reduction and three had no change during post-testing. Five individuals displayed a reduction in braking duration. One increased duration and five had no change. Five participants saw an increase in braking impulse, four had a reduction and two had no change. Nine participants increased propulsive mean force with two experiencing no change (Figure 1). Six individuals saw a reduction in propulsive duration with three increasing duration. Two participants had no change in propulsive duration (Figure 2). Propulsive net impulse was increased in 10 participants and no change was seen in one (Figure 3). Countermovement depth was reduced in three participants, increased in two, and had no change in six. Jump height increased in ten participants and one had no change. Time to takeoff was reduced in seven individuals, increased in two individuals, and showed no change in two individuals. Lastly, RSIm was increased in nine individuals and no change was seen in two.

## 4. Discussion

The main findings of this investigation were that, in the NAS condition, jump performance saw more changes from pre- to post-practice than the number of changes seen in the AS condition. These findings both support previous investigations, where no change was seen in the CMJ performance of volleyball athletes after fatiguing tasks, and are in contrast to the greater body of literature on changes in CMJ performance as a result of fatigue [12,14,23,24]. In a review of changes in physical performance testing post-competition, CMJ jump height was reduced between 1.6 and 6 cm [23]. Within this review, a variety of sports were used, and testing occurred at a variety of time intervals post-competition. While the general trend of a reduction in jump height was seen, no controls for the type of CMJ or how the CMJ was measured were used in the review [23]. Thus, a wide range of reductions in jump height and effect sizes (0.22–1.22) were seen [23]. Gathercole et al. [4] previously displayed that different forms of vertical jump testing in a fatigued state provided different results. Under the same fatigue conditions, countermovement jump performance displayed different findings than squat jumps in terms of fatigue sensitivity [4]. This illustrates similar findings to the present investigation where the NAS condition displayed changes in jump height that were not seen in the AS condition. Not only is it important to select which vertical jump assessment is used, but other methodological considerations, such as arm swing utilization, also need to be accounted for.

Previous investigations have used either the AS or NAS jump test based on a variety of factors. If using the Vertec device to assess jump performance then an AS has to be employed, whereas if using linear transducers, force plates, or jump mats, either methodology can be used. This creates potential issues in the literature as it has been shown that within-subject outputs and jump strategies can shift based on the use of an AS [7,8,9,25]. Hoffman et al. found [13] there to be no differences pre- and post-competition when completing a NAS testing protocol in collegiate female soccer athletes, whereas McLellan et al. [26] found there to be a decrease in peak force in post-match testing of rugby league athletes using an AS. This lack of consistency in findings throughout the literature can be explained through a variety of factors concerning methodologies. The findings in the current study point to the need for consistency in the literature as a change from pre- to post-testing differed based on the jump condition used. This is the first investigation, to the author’s knowledge, that used both AS and NAS conditions to assess the changes to jump performance in a fatigued state. Previous investigations that have used multiple forms of vertical jump have manipulated the countermovement itself by using the squat jump (SJ) or depth jumps [4]. Gathercole et al. [4] found that the CMJ task was best in assessing immediate and prolonged neuromuscular fatigue in the jumping task used. Interestingly, CMJ performance was diminished during the immediate post-exercise condition using a NAS methodology where performance was improved in the current study [4]. The results of the current study, however, support the findings of Moreno-Perez et al. [27] who found an increase in jump height post competition in semi-professional basketball athletes. This coincided with an increase in the dorsiflexion range of motion. Additionally, in a sample of snowboard athletes, a small increase in jump height post excise was observed [28]. This increase in jump height occurred with a decrease in propulsive mean force and an increase in propulsive time as well as time to takeoff [28]. This is in line with the current study, where an increase in propulsive mean force increased and time to takeoff decreased, suggesting a change in movement strategy during the immediate fatigued state.

As has been stated in many of the previous investigations centered on jump testing to assess neuromuscular fatigue, the source of the fatigue (exercise, sport-specific training, competition) and the athletes themselves, play a critical role in the findings of this study. Cooper et al. [24] found there to be a significant reduction in jump height, force, and power in a sample of recreationally trained individuals after completing a fatiguing task of continuous vertical jumps. However, Robineau et al. [29] found there to be no statistical reduction in CMJ height after a simulated soccer game in eight amateur soccer athletes. Moreover, Cortis et al. [30] found an increase non-statistically significant increase in jump height in a sample of 10 senior male soccer athletes after a match. This demonstrates that sample demographics and the method by which fatigue is induced can impact findings related the vertical jump performance. In a similar study to the current investigation, professional male volleyball athletes displayed no differences in any CMJ metric after a sport-specific training session [15]. However, the authors failed to report whether individuals were allowed to use an AS. Though the exact methods used were not disclosed, the results are similar to the present investigation. As volleyball is a sport that relies heavily on vertical jump ability, changes in jump performance may be limited. With volleyball athletes needing to complete the vertical jump task in fatigued states during training and competition, post-testing may not produce significant changes. Based on the previous findings and those of the current investigation, future investigations should examine the changes in CMJ metrics over consecutive days of training and competition [15].

The use of the single-subject analysis in this study provides additional valuable information that has not previously been reported. A general trend was seen across all participants during the NAS that was not seen during the AS condition. An example of this can be seen in the increase in propulsive mean force, where nine of the eleven subjects saw an increase in the NAS condition. During the AS condition, propulsive mean force increased in four participants, reduced in four, and had no change in three. This is important for several reasons. First, of the four individuals displaying an increase in AS propulsive mean force, two had an increase in propulsive net impulse during the AS condition; both of these individuals also showed increased propulsive duration. The other two participants that saw an increase in propulsive mean force and had reductions in propulsive duration resulting either in no change or a reduction in propulsive net impulse. This indicates a potential change in the strategy being used that is masked at the group level by individuals having the opposite strategy shift occur (reduce force and increase duration). Thus, practitioners interested in changes in performance as a result of competition and practice should use single-subject analysis rather than group means, as individuals can respond to similar exercises and stresses differently.

This study is not without limitations. First, the sample size used for this investigation is small. This is due to the roster size during the spring training period (offseason) in which new members of the team and injured athletes were not taking part in training as they would during the competitive season. However, though the sample size is small, previous investigations have used similar sample sizes [15,29,30,31] Secondly, this investigation was conducted using a cross-sectional design and results from this study may not be transferable across all training sessions. The training load based on individual jump counts is similar to what has been previously reported in professional volleyball athletes [32]. Based on the previous investigations that have examined pre- and post-changes based on a fatiguing exercise session or game scenario, we feel that the outcomes of this investigation provide a unique examination of jump performance change on both the group and individual levels [13,15,26].

## 5. Conclusions

In conclusion, CMJ performance post-sport-specific training appears to be influenced by the jump testing methodology used. This is important for practitioners to consider when selecting a methodology to assess neuromuscular fatigue in their athletes. As both the NAS and AS conditions displayed changes on the individual level, practitioners using both methodologies appear suited for assessing neuromuscular fatigue. Practitioners should also consider examining individual changes in addition to group changes as every variable in the current study is subject to a level of individual change.

## Figures and Tables

**Figure 1 jfmk-08-00137-f001:**
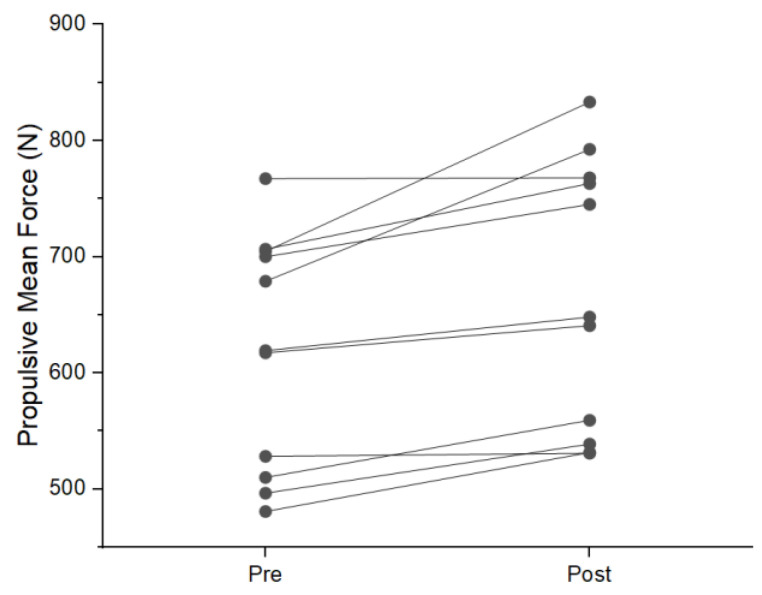
Individual change in propulsive mean force during NAS condition.

**Figure 2 jfmk-08-00137-f002:**
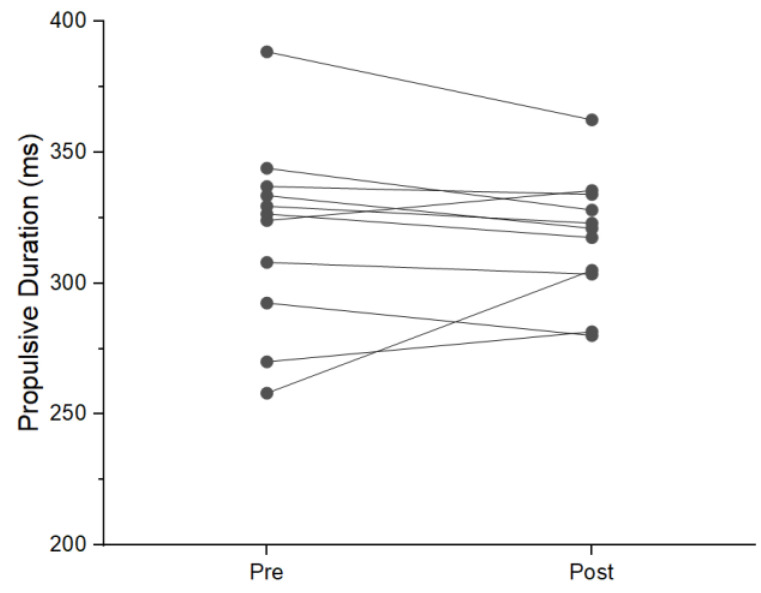
Individual change in propulsive duration during NAS condition.

**Figure 3 jfmk-08-00137-f003:**
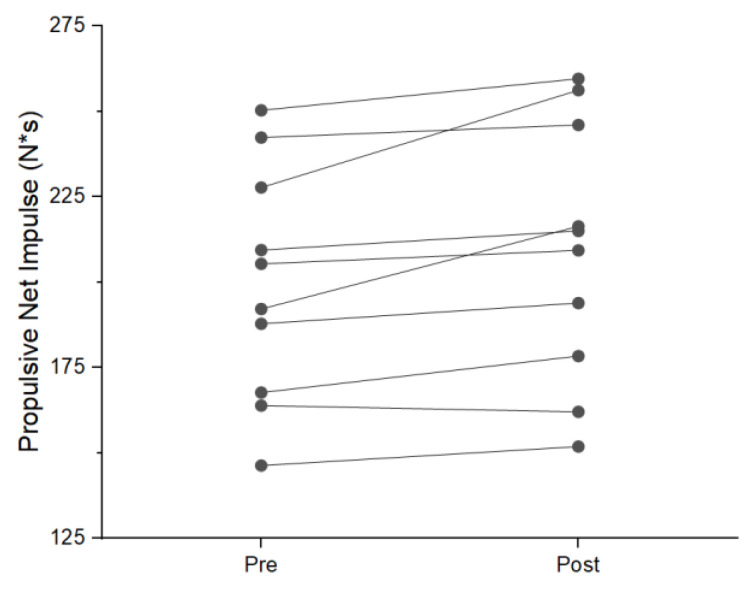
Individual change in propulsive net impulse during NAS condition.

**Table 1 jfmk-08-00137-t001:** Intraclass Correlation Coefficient (ICC) and Coefficient of Variations (CV).

	ICC (95% CI)	CV (95% CI)
Arm Swing		
Braking Mean Force	0.91 (0.77–0.96)	3.71 (1.49–5.93)
Braking Duration	0.94 (0.86–0.97)	5.22 (2.01–8.42)
Braking Impulse	0.84 (0.61–0.93)	5.77 (2.75–8.79)
Propulsive Mean Force	0.99 (0.97–0.99)	1.81 (0.76–2.86)
Propulsive Duration	0.86 (0.64–0.95)	2.12 (1.03–3.22)
Propulsive Net Impulse	0.97 (0.89–0.99)	1.84 (0.37–3.30)
Countermovement Depth	0.88 (0.61–0.96)	3.26 (2.09–4.42)
Time To Takeoff	0.94 (0.87–0.98)	1.08 (0.55–1.61)
Jump Height	0.97 (0.93–0.99)	1.97 (1.14–2.80)
RSIm	0.95 (0.90–0.98)	2.39 (1.30–3.48)
No Arm Swing		
Braking Mean Force	0.91 (0.78–0.96)	6.38 (2.92–9.81)
Braking Duration	0.93 (0.86–0.97)	5.89 (3.22–8.56)
Braking Impulse	0.84 (0.61–0.93)	4.16 (3.03–9.76)
Propulsive Mean Force	0.91 (0.80–0.96)	2.79 (1.53–4.05)
Propulsive Duration	0.88 (0.80–0.95)	2.90 (1.66–4.14)
Propulsive Net Impulse	0.93 (0.85–0.97)	1.62 (0.44–2.80)
Countermovement Depth	0.88 (0.61–0.96)	3.56 (2.03–5.08)
Time To Takeoff	0.94 (0.87–0.98)	2.17 (1.03–3.30)
Jump Height	0.97 (0.93–0.99)	2.74 (0.87–4.61)
RSIm	0.95 (0.90–0.98)	4.10 (2.16–6.04)

**Table 2 jfmk-08-00137-t002:** Changes from Pre to Post Testing (mean ± SD).

Arm Swing Condition					
	Pre	Post	*p*	*g*	%Δ
**Mean Braking Force (N)**	**496.93 ± 145.62**	**536.29 ± 170.51**	**0.047**	**0.656**	**7.9**
Braking Duration (ms)	201.41 ± 39.39	196.86 ± 56.70	0.704	0.114	2.6
Braking Net Impulse (N*s)	96.46 ± 24.03	100.31 ± 28.52	0.386	0.263	4.0
Mean Propulsive Force (N)	624.42 ± 119.76	630.52 ± 119.91	0.413	0.248	1.0
Propulsive Duration (ms)	336.59 ± 23.21	339.91 ± 35.86	0.573	0.169	1.0
Propulsive Net Impulse (N*s)	210.27 ± 35.16	213.19 ± 28.73	0.554	0.178	1.4
Countermovement Depth (cm)	39.51 ± 5.66	41.08 ± 7.11	0.103	0.521	4.0
Jump Height (cm)	33.84 ± 4.74	34.19 ± 4.46	0.545	0.182	1.0
Time to Takeoff (ms)	992.13 ± 99.26	971.95 ± 89.05	0.321	0.303	2.0
RSImod	0.34 ± 0.05	0.35 ± 0.05	0.133	0.475	2.9
No Arm Swing Condition					
	**Pre**	**Post**	** *p* **	** *g* **	**%Δ**
Mean Braking Force (N)	527.45 ± 155.94	570.61 ± 149.25	0.097	0.551	8.2
Braking Duration (ms)	189.23 ± 34.28	170.73 ± 42.92	0.099	0.527	9.8
Braking Net Impulse (N*s)	95.89 ± 20.73	92.54 ± 19.63	0.548	0.188	3.5
**Mean Propulsive Force (N)**	**619.03 ± 100.74**	**668.30 ± 116.16**	**0.002**	**1.179**	**8.0**
Propulsive Duration (ms)	319.23 ± 36.23	317.41 ± 24.14	0.764	0.089	0.6
**Propulsive Net Impulse (N*s)**	**197.81 ± 33.04**	**213.19 ± 38.67**	**0.038**	**0.695**	**7.8**
Countermovement Depth (cm)	35.90 ± 6.39	35.58 ± 6.16	0.421	0.244	0.9
**Jump Height (cm)**	**28.95 ± 4.97**	**31.27 ± 5.00**	**0.001**	**1.700**	**8.0**
**Time to Takeoff (ms)**	**873.09 ± 103.28**	**831.95 ± 99.83**	**0.015**	**0.852**	**4.7**
**RSImod**	**0.33 ± 0.06**	**0.38 ± 0.07**	**0.001**	**1.466**	**15.2**

**Bold** values represent statistically significant differences between time points. RSImod = reactive strength index modified.

## Data Availability

The data presented in this study are available on request from the corresponding author.

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
