# Peer review of "Examination of Countermovement Jump Performance Changes in Collegiate Female Volleyball in Fatigued Conditions"

_jfmk, 2023, doi:10.3390/jfmk8030137_

Round 1

Reviewer 1 Report

The main problem of the article is the sample. I think it is a very small number of subjects for the results to have significant significance. On the other hand, the subject is not very original and does not have much bibliographical foundation.

Author Response

The main problem of the article is the sample. I think it is a very small number of subjects for the results to have significant significance. On the other hand, the subject is not very original and does not have much bibliographical foundation.

Thank you for taking the time to review our manuscript. We have made several adjustments to the discussion section to try and make this manuscript more suitable.

As for the sample size we agree that this is small and a concern. We addressed this concern in the last paragraph of the discussion and have provided citations demonstrating that unfortunately, when doing ecological valid research with a group of athletes we are limited by the size of the roster and the number of healthy athletes. We appreciate your concern.

Reviewer 2 Report

This study aimed to examine changes in countermovement vertical jump performance after a single sport-specific training session in a sample of collegiate female volleyball athletes. This is a well-written paper which adds knowledge to the related-fatigue based on CMJ (with and without arms) along training sessions in volleyball. My major concern, as I will explain, is about a unique training session analyzed. Moreover, I have some other minor points:

The following studies could be really to include in the final paragraph of introduction and in some places of discussion:

·       Castillo, D, Yanci, J, & Cámara, J (2018). Impact of official matches on soccer referees’ power performance. Journal of Human Kinetics, 61, 131-141. 

·       Claudino, J. G., Cronin, J., Mezêncio, B., McMaster, D. T., McGuigan, M., Tricoli, V., Amadio, A. C., & Serrão, J. C. (2017). The countermovement jump to monitor neuromuscular status: A meta-analysis. Journal of Science and Medicine in Sport, 20(4), 397–402.

·       Cortis C, Tessitore A, Lupo C, Pesce C, Fossile E, Figura F, Capranica L (2011). Inter-limb coordination, strength, jump, and sprint performances following a youth men’s basketball game. Journal of Strength and Conditioning Research, 25, 135-142.

Again, I want to focus on my major concern. Just one training session analyzed, performing pre- and post-, CMJ with and without arms. In this sense, it would be interesting to know the inter-session variability. This aspect could change the findings of the authors. It could be that the pre- and post- CMJ results are different in other testing sessions.

I suggest authors to perform the statistical power analysis. For example, here: https://www.psychologie.hhu.de/arbeitsgruppen/allgemeine-psychologie-und-arbeitspsychologie/gpower

Lines 67-68. Include units of age, height and body mass.

Line 72. Include code of Ethics Committee.

Another important point is the procedures. In this section, authors should explain if the volleyball players performed a familiarization period to learn the CMJ technique. Please, provide duration, frequency, etc.

I think it is very appropriate that the authors include a figure explaining the CMJ jumps: NAS and AS.

Other important aspect is to explain the volume and intensity of the training session. In addition, include the physical and psychophysiological demands of this session.

How authors calculate the ICC and CV? At pre? At post? Both?

It would be advisable to link tables 2 and 3 in a unique table and add percentage of change.

Lines 172-174. “The main findings of this investigation were that in the NAS condition jump performance changed to a greater extent than the AS condition after a session of sport-specific training in female collegiate volleyball athletes.” Authors cannot conclude this statement because the statistical analyses do not allow to compare between NAS and AS.

Line 207. Here, why do authors think that there is an improvement? Also, authors should add a practical application.

Lines 208-222. I encourage authors to delete this paragraph. This is a simple example.

Lines 223-234. This paragraph should not be appropriated. Some of this information should be included in paragraph 2 of the discussion section.

Finally, I have some doubts with the conclusion. Why do authors that NAS is better to analyze training related-fatigue? Could authors define the practical applications of the study?

Reviewer 3 Report

The purpose of the study was to examine changes in countermovement vertical jump performance (CMJ) after a sport-specific training session in female volleyball athletes. A total of 11 college volleyball players performed. CMJ was quantified both with and without arm swing before and after a sport specific volleyball practice session. The main findings were that during are swing conditions only the mean breaking force out of all the dependent variables was changed. In the arm swing condition mean propulsive force, propulsive net impulse, jump height and reactive strength index demonstrated significant increases whereas time to takeoff was decreased.

 The strengths were: trained volleyball population used, methods of data collection appeared to be sound, well-written with few typos and grammar errors, stats well described and presented, and relatively good job acknowledging limitations.

 One weakness was relatively low sample size (but this is understandable and was acknowledged.)

 Unfortunately, some major interrelated weaknesses exist:

1)      no variables indicative of the total workload of the practice were taken, which I assume has probably taken place in prior studies. Therefore, how can this study be compared to other studies that did or didn’t do the same. How fatigued could the authors estimate they really were if they didn’t measure any variables related to the general volume or intensity of the practice (e.g. total jumps, total steps, average heart rate, or many other possible variables). We were only told it was a 2 hour sports specific practice.

2)      Relatedly, the authors assumed some level of fatigue. But fatigue in the classic sense wasn’t measured (the percentage decline in MVC force). Although of course the authors did measure power related measures that could be indices of fatigue. However, it would have been ideal to measure MVC force in an isometric leg extension or squat before and after. This is a more minor issue than number 1 above.

 Thus, for the study to be publishable I think the authors need to address these 2 major weaknesses and explain why these are not fatal flaws of the study that make the data uninterpretable. If this can be done then the study can likely be publishable.

 Minor comments:

-Line 25, no period after [3]

Author Response

Thank you for taking the time to review our manuscript and provide the insightful comments.

  • No variables indicative of the total workload of the practice were taken, which I assume has probably taken place in prior studies. Therefore, how can this study be compared to other studies that did or didn’t do the same. How fatigued could the authors estimate they really were if they didn’t measure any variables related to the general volume or intensity of the practice (e.g. total jumps, total steps, average heart rate, or many other possible variables). We were only told it was a 2 hour sports specific practice.

Thank you for bringing this up. We have obtain data from the sport coaches regarding this session and have added the following.

During the training session, six participants wore inertial sensors (Vert, Mayfonk Athletic, Florida USA) to measure jump counts. The average jump count for the six participants during the session was 103.66 with a range of 81 to 165 jumps.

  • Relatedly, the authors assumed some level of fatigue. But fatigue in the classic sense wasn’t measured (the percentage decline in MVC force). Although of course the authors did measure power related measures that could be indices of fatigue. However, it would have been ideal to measure MVC force in an isometric leg extension or squat before and after. This is a more minor issue than number 1 above.

Thank you again for the insightful comment. We completely understand were you are coming from with the MVC data. That would have been ideal. Unfortunately for the purposes of this particular investigation, all testing was completed at the court the team utilizes which does not have access to any MVC testing where through leg extensions or the use of an isometric squat.

 Thus, for the study to be publishable I think the authors need to address these 2 major weaknesses and explain why these are not fatal flaws of the study that make the data uninterpretable. If this can be done then the study can likely be publishable.

Round 2

Reviewer 2 Report

I think authors have improved the manuscript. However, I feel more comfortable whether they add some references after this sentence: "Based on the previous investigations that have examined pre and post- changes based on a fatiguing exercise session or game scenario".

Again, I encourage authors to perform a posteriori statistical power analysis. Also, include the results in participants section, please.

Regarding Ethics Committee. Which University? Please, add name and code.

Author Response

Again we would like to thank the reviewer for their time providing comments regarding our submission. Please find our responses below.

I think authors have improved the manuscript. However, I feel more comfortable whether they add some references after this sentence: "Based on the previous investigations that have examined pre and post- changes based on a fatiguing exercise session or game scenario".

The following citations have been added to the end of this sentence.

Based on the previous investigations that have examined pre and post-changes based on a fatiguing exercise session or game scenario, we feel that the outcomes of this investigation provide a unique examination of jump performance change on both the group and individual levels [13,15,26]

Again, I encourage authors to perform a posteriori statistical power analysis. Also, include the results in participants section, please.

A post-hoc power analysis was performed using G*Power (version 3.1.9.7). This calculation has completed using the jump height from the no arm swing condition (NAS) with an effect size of 1.76. Observed power was calculated as 0.99.

Regarding Ethics Committee. Which University? Please, add name and code.

This study was conducted according to the guidelines of the Declaration of Helsinki and was approved by the University of Southern Mississippi institutional review board (20-478). 

Reviewer 3 Report

The authors have answered all of my previous comments. I think the paper can be published.

Author Response

Thank you again for taking the time to review our manuscript submission.